# The Effect of Low-Modulus Plastic Fiber on the Physical and Technical Characteristics of Modified Heavy Concretes Based on Polycarboxylates and Microsilica

**DOI:** 10.3390/ma15072648

**Published:** 2022-04-04

**Authors:** Daniyar Akbulatovich Akhmetov, Yuri Vladimirovich Pukharenko, Nikolai Ivanovich Vatin, Sungat Berkinovich Akhazhanov, Akbulat Raiymbekovich Akhmetov, Ainur Zhenisbekkyzy Jetpisbayeva, Yelbek Bakhitovich Utepov

**Affiliations:** 1NIISTROMPROJECT (LLP), Almaty 050000, Kazakhstan; dan-akhmetov@yandex.kz (D.A.A.); tsik@spbgasu.ru (Y.V.P.); stjg@mail.ru (S.B.A.); 2Department of Building Materials and Metrology, Saint Petersburg State University of Architecture and Civil Engineering, 190005 Saint Petersburg, Russia; 3Higher School of Industrial, Civil and Road Construction, Peter the Great Saint Petersburg Polytechnic University, 195251 Saint Petersburg, Russia; vatin@mail.ru; 4Faculty of Mathematics and Information Technology, Karaganda Buketov University, Karaganda 100024, Kazakhstan; 5Department of Industrial, Civil and Road Construction, M. Auezov South Kazakhstan University, Shymkent 160012, Kazakhstan; aakbulat@mail.ru; 6Department of Construction and Building Materials, Satbayev University, Almaty 050013, Kazakhstan; 7Department of Civil Engineering, L.N. Gumilyov Eurasian National University, Nur-Sultan 010008, Kazakhstan

**Keywords:** durability, modifiers, hyperplasticizers, fibers, heavy concrete, microsilica

## Abstract

Manufacturers of building materials strive to optimize the three basic concrete properties—strength, durability, and shrinkage deformation, of which the focus is generally on the durability in the structure when designing and monitoring the poured concrete. Studying concretes’ structural performance and the change in their characteristics over time enables the solution of many important issues associated with the design of reliable, durable, and cost-effective buildings and structures. This article presents studies aimed at improving the physical, technical, and operational characteristics of cement concrete and reducing cement consumption in heavy concretes through the use of complex modifiers and volumetric fiber reinforcement. Four concrete compositions of widely recognized grades were developed, of which samples were molded and tested for compressive and flexural strength, frost resistance, volumetric water absorption, and density. Test results confirmed the possibility of binder (cement) economy up to 18% and increasing frost resistance up to W300 when using microsilica, reduction in volumetric water absorption of up to 40% when using both microsilica and hyperplasticizer, and increasing flexural strength by over 30% when using polymer fiber. The developed compositions passed the industrial tests, and were successfully introduced in the production process of the operating reinforced concrete products’ manufacturer.

## 1. Introduction

In global practice, the production of high-strength quick-hardening concretes is increasing, which allows the development of structural elements and technologies, significantly expanding the assortment of concrete products and structures [1,2]. From the analysis of the literature [3,4] on this subject, it follows that little attention is paid to the issue of reducing the proportion of cement in the composition of concrete. This important issue is dictated not only by economic but also by environmental aspects, as huge emissions of carbon dioxide (CO_2_) occur during the firing of carbonate raw materials in the production of Portland cement [5]. Moreover, little attention is paid to assessing the durability and serviceability of products made of heavy concrete [6,7]. In practice, although not always, manufacturers use only the method of determining frost resistance [8], used both when selecting the composition of concrete and in the process of quality control of concrete products and structures. The development of modified concretes, considering reducing their cost by decreasing the amount of Portland cement in the composition of concrete and improving their quality and durability in relation to operating conditions, is topical [3]. Complex studies to improve physical and technical characteristics, in terms of assessment of frost resistance indicators, allow objective estimation of the service life of modified concrete. Similar characteristics also include the bending strength and indicators of operational durability and longevity. The latter include assessment by volumetric water absorption, as well as assessment of meteorological features of the region (transition through the zero-degree mark during the year and ambient humidity) [9,10]. These developments enable design and construction organizations to use modified concretes expediently in construction.

Effective heavy concretes of various functional purposes with improved building and operating properties are produced using multicomponent compositions [11]. The creation of such concretes is based on the principle of purposeful management of technology at all of its stages: the use of active components [1]; the development of optimal compositions [12]; the use of chemical modifiers; and reducing macroporosity and increasing cracking resistance by strengthening the contact zones of the cement stone and coarse aggregate [13]. Previously published papers [14] show that with the introduction of ultradispersed silica waste to the quick-hardening concrete, in combination with a hyperplasticizer, it is possible to obtain concrete with strength of up to 100 MPa after 1 day of normal curing, and with the use of hardening accelerators up to 20 MPa after 4 h and 130 MPa after 7 days of curing.

Active mineral additives significantly affect the basic physical and technical properties of concrete, giving it improved characteristics contributing to more efficient use of chemical energy from clinker by creating additional centers of crystallization. The number of stable hydrosilicates increases due to the reduction in the most unstable component of cement stone—Ca(OH)_2_ crystals—which is important for the manufacture of dense and durable concrete [15]. Analysis of studies of the processes of structural formation of multicomponent cement systems shows that the activity of mineral additives is characterized by their ability to exert both chemical and physicochemical effects on cement hydration processes. The chemical activity of the studied ferroalloy slag waste—i.e., microsilica—is mainly pozzolanic in nature [16]. The surface energy can serve as a criterion for both chemical and physicochemical activity of the studied mineral additives. According to the thermodynamic concept of adhesion, the main role in forming adhesive strength is assigned to the ratio of the significant surface energy of the adhesive and the substrate [17]. To obtain high-strength quick-hardening concretes, it is proposed to reduce the pore size by introducing ultradispersed active silica as a waste product of ferrosilicon production [18].

One of the promising ways to improve the physical, technical, and operational characteristics of concrete is the introduction of polymeric fiber [19,20]. The use of this material can increase the tensile strength of the cement matrix, allowing a cement concrete of higher flexural strength to be obtained [21]. It is recommended to use polypropylene fiber in the technology of fiber concrete to reduce the delamination of mixtures, increase water resistance, frost resistance, corrosion resistance, and impact strength, and reduce abrasion [2,22]. In addition, polypropylene fiber is characterized by corrosion resistance in various environments [23]. The economic aspect of research relevance was also considered when choosing this type of fiber for research, since the polymer fiber is much cheaper than metal fiber for the same volume [24].

Analysis of the aforementioned literature and the experience of the authors of this article—accumulated over several years of inspection work at the plants for the production of reinforced concrete products, along with systematic observations of the products during operation—show that it is very important not only to obtain concrete with highly desirable characteristics, but also to maintain those characteristics during the period of operation [24]. According to the authors, in order to obtain high-quality modified concretes with good performance characteristics, the following is necessary:To improve the technological properties of concrete mixtures by reducing their water content, improving their workability, and increasing their homogeneity and resistance to stratification, by using a complex modifier consisting of a hyperplasticizer based on polycarboxylate ester and ultradispersed active silica;To reduce the shrinkage deformations of concrete in the early stage of curing and increase the flexural strength of cement concrete through the use of volumetric fiber reinforcement with low-modulus polypropylene fiber, in order to improve the crack resistance and quality of products.

The purpose of this study was to improve the physical–technical and operational characteristics of cement concrete and reduce cement consumption of mainstream grades of heavy concrete by using complex modifiers and volumetric fiber reinforcement. This goal was achieved by using complex multicomponent compositions and obtaining high-quality concretes of different functional purposes with improved building and operational properties.

The main technical task of this study was to reduce the proportion of cement in the composition of concrete. This necessity is dictated not only by the reduction in energy intensity in the production of Portland cement and the reduced consumption of fuel and electricity in the extraction, processing, and handling of raw materials, but also by the most pressing issue—significant carbon dioxide emissions during the firing of carbonate raw materials. This task was solved based on the theory of creating additional crystallization centers by using reactive pozzolanic additives (75–95% active SiO_2_) that bind hydrolyzed lime to additional amounts of strong cementitious calcium hydrosilicates. In parallel, by creating additional crystallization centers, the issue of increasing the durability of concrete was addressed.

The next technical task was to reduce the water–cement (W/C) ratio and increase early and final strength indicators of concrete, increasing frost resistance and reducing the volumetric water absorption index of concrete while maintaining a given flowability, according to the basic law of concrete strength [25], through the use of polycarboxylate-ester-based high-water-reducing additives.

The task of enhancing the deformation properties and cracking resistance of concrete by using dispersed polypropylene fiber reinforcement was the final task of this study. Here, of particular interest was the dependence of the effect of fiber on the compressive strength and flexural strength of popular grades of heavy concrete with different binder consumption and water–cement ratios.

## 2. Materials and Methods

This study used both theoretical and empirical research methods. Theoretical research aimed to calculate the compositions of modified heavy concrete using the equation of absolute volumes and compare the results with existing compositions of heavy concrete of popular grades, determining the direction of the work, focusing on cement saving. Empirical studies were aimed at experimental confirmation of theoretically developed compositions. Furthermore, studies of physicomechanical and operational properties of the obtained modified concrete of popular grades were carried out. The studies were carried out in 4 stages, each of which was aimed at solving a specific task:Selection of compositions of modified heavy concrete of popular grades, as described in [26];Determination of the basic strength characteristics (compressive and flexural strength) at the design age of 28 days;Determination of basic performance characteristics (i.e., frost resistance, volumetric water absorption, average density);Comparison and analysis of test results of modified heavy concrete of mainstream grades.

This study used local raw materials. Portland cement CEM I 52.5 N manufactured by Heidelberg Cement (Heidelberg, Germany) in Kazakhstan was used as a binder for the studied concrete mixtures. Several tests were carried out to confirm the compliance of the chosen binder with the regulations and requirements of [27]. The methods specified in the standard were used to determine the following parameters: grinding fineness; normal density and setting time of the cement dough; and compressive and flexural strength.

Thus, the investigated binder showed a grinding fineness of 97.4%. Its normal density was 26.3%. The beginning of the setting was in 2 h 11 min, while the end of the setting was in 4 h 10 min from the moment of mixing. The values obtained were compliant with the standard. In determining the strength characteristics, the binder under study showed a bending strength of 6.1 MPa at the age of 28 days, in compression at 53.7 MPa. The obtained parameters corresponded to the standard.

Sand from the manufacturer Giyada LLP (Almaty, Kazakhstan), meeting the standard of [28], was used for the tests. According to the standard set out in [28], sands with a maximum amount of dust and clay inclusions of 3% for the groups increased coarseness can be used as a fine aggregate for heavy concrete. However, according to [24], to obtain satisfactory characteristics of the concrete mixture and the final conglomerate, it is necessary to use sand with dust inclusions of less than 1.5%. According to the test results, the content of dust and clay inclusions in the studied sand was 1.08%. Furthermore, according to [28], by sieving and determining the grain composition of the aggregate, the coarseness modulus of the studied sand was determined, and amounted to 2.6, which corresponds to a size fraction of 0.16–5 mm. These values are acceptable for using the studied aggregates in heavy concrete.

Crushed stones of 5–10 mm and 10–20 mm fractions produced by Baltabay LLP (Almaty, Kazakhstan), with known physical and technical characteristics, were used in the coarse aggregate. This aggregate meets the requirements of [29], which defines the basic requirements for crushed stone from dense rocks used as an aggregate for heavy concrete, including self-compacting concrete (SCC).

Condensed compacted microsilica CCM-85 manufactured by Tau-Ken Temir LLP (Karaganda, Kazakhstan) was used as a reactive pozzolanic additive, as described by [30].

A chemical additive based on polycarboxylate esters of the 2nd generation AR Premium described by [26], produced by ARPG LLP (Nur-Sultan, Kazakhstan), with the characteristics listed in Table 1, was used as a hyperplasticizer (HPC).

Low-modulus polypropylene fiber of different lengths from Fibro-Lux LLC (St. Petersburg, Russia) was used for volumetric reinforcement in the studies, as described by [31], with the following characteristics (Table 2).

The structure of further studies was to compare the indicators of physical–technical and performance characteristics of the modified concrete of popular grades, obtained by mathematical calculation using the equation of absolute volumes, and experimentally confirmed by laboratory mixes and studies of physical–technical and performance characteristics at the design age of 28 days. All studies and tests were conducted following interstate standards [32,33,34,35,36,37].

### 2.1. Calculation of the Concrete Composition of Popular Grades

The calculation involved four steps:Formulation of technical specifications which, in addition to the conventional requirements for concrete strength and workability of the concrete mix, must contain the minimum allowable value of its ultimate extensibility, based on the conditions of formation and, therefore, the expected or measured value of the possible deformations;Selection of raw materials and obtaining the necessary data characterizing their properties, including the water consumption of aggregates;Calculation of the concrete mix composition to determine the consumption of key components in kg per 1 m^3^;Experimental verification of the obtained composition and its adjustment (if necessary).

The concrete compressive strength *R_b_* was determined according to its dependence on the cement activity *R_c_* and the *W*/*C* ratio
(1)$$R_{b}=\frac{R_{c}}{k{\left(\frac{W}{C}\right)}^{n}}$$
where *k* and *n* are coefficients depending on the type of concrete and the quality of aggregates, respectively. For heavy concrete, *n* = 1.5; when using crushed stone, *k* = 3.5; for gravel, *k* = 4. This equation is valid for calculating the strength of densely placed concrete, cured under normal temperature and humidity conditions and tested by the standard method at the age of 28 days.

Next, the consumption of coarse and fine aggregates (kg/m^3^) was determined using Equations (2) and (3). The sum of the consumption of components is numerically equal to the average density of the concrete mixture, expressed in kg/m^3^. The calculation of the composition was based on solving a system of equations for fine-grained (Equation (4)) or heavy (Equation (5)) concrete, which include mathematical expressions that are already known and widely used in practice, such as the basic law of concrete strength, the equation of absolute volumes, and the separation factor of aggregate grains, which together can provide the relationship of consumption and the properties of the main components with the concrete strength and workability of the concrete mixture.
(2)$$CS=\frac{1000}{\frac{V_{h.CS}\cdot k_{\rho }}{\rho_{bd.CS}}+\frac{1}{\rho_{nd.CS}}}$$
(3)$$S=\left[100-\left(\frac{C}{\rho_{C}}+\frac{W}{\rho_{W}}+\frac{CS}{\rho_{S}}\right)\cdot \rho_{S}\right]$$
(4)$$\{\begin{array}{c} R_{b}=R_{c}\cdot A\cdot \left(\frac{C}{W}\pm 0.3\right)\\ \frac{W}{\rho_{W}}+\frac{C}{\rho_{C}}+\frac{S}{\rho_{S}}+F=1 \end{array}$$
(5)$$\{\begin{array}{c} R_{b}=R_{c}\cdot A\cdot \left(\frac{C}{W}\pm 0.5\right)\\ \frac{W}{\rho_{W}}+\frac{C}{\rho_{C}}+\frac{S}{\rho_{S}}+\frac{CS}{\rho_{CS}}+F=1\\ r=\frac{S}{CS} \end{array}$$
where:

*Rb* is the given concrete strength (MPa);

*V_h_._CS_* is the hollowness of crushed stone in bulk condition;

*k_p_* is the coefficient of aggregate grain separation;

*p_bd_._CS_* is the bulk density of crushed stone;

*ρ**_nd_._CS_* is the natural density of crushed stone; 

*Rc* is the cement activity (MPa); 

*A* is the coefficient counting the quality of aggregates; 

*C*, *W*, *S*, *CS*, and *F* are the consumption of cement, water, sand, crushed stone, and fiber, respectively, per 1 m^3^ of concrete (kg) (fiber consumption is determined by indicators of workability by test batches);

*ρ**_C_*, *ρ**_S_*, and *ρ**_CS_* are the grain density of cement, sand, and crushed stone, respectively (kg/m^3^);

*r* is the ratio of sand and crushed stone consumption, which is assigned based on the grain composition of aggregates (for a crushed stone fraction of 5–20 mm, the recommended *r* = 0.55–0.8; for a crushed stone of fraction of 5–10 mm, the recommended *r* = 0.65–1.5).

Table 3 shows summary data of the concrete compositions obtained by the calculations above.

The concrete compositions were then experimentally validated during laboratory batching, after which control samples of concrete were made and tested for the corresponding characteristics.

### 2.2. Determination of Compressive and Flexural Strengths

As part of the compression and tensile tests, specimens were molded in the forms 2FK 100 and FP 100, according to the standard in [33], from the mixture of the control composition, and then from each subsequent test composition from Table 3. Furthermore, when the specimens reached the design age of 28 days, tests were carried out according to the standard in [34] (Figure 1).

The flexural strength of concrete was calculated with an accuracy of 0.01 MPa using Equation (6):(6)$$R_{bt}=\delta \frac{Fl}{ab^{2}}K_{W}$$
where: 

*F* is the breaking load (N);

*a*, *b*, and *l* are the width and height of the cross-section of the prism beam, and the distance between the supports, respectively, in the flexural test of samples (mm);

*δ* is the scaling factor for converting the strength of concrete to the strength of concrete in samples of basic size and shape; 

*K_W_* is the correction factor for cellular concrete, taking into account the moisture content of samples during the test (not applicable for heavy, fine-grained, or SCC).

After obtaining data on the compressive strength and calculations of the flexural strength of samples, the dependence of the influence of modifiers and volumetric fiber reinforcement on the flexural strength was determined.

### 2.3. Determination of the Frost Resistance

The test to determine the frost resistance was carried out via the second accelerated method described in [8]; the samples were frozen in the air, saturated with sodium chloride, and thawed in a mixture of sodium chloride. Accelerated tests were carried out under the conditions given in Table 4 and Table 5.

After obtaining data on frost resistance, the dependence of the effect of modifiers and volumetric fiber reinforcement on the concrete frost resistance was defined.

### 2.4. Determination of Water Absorption

Tests to determine water absorption in concrete were conducted as described in [35], where concrete samples were placed in a container filled with water so that the water level in the container was above the top level of the placed samples by about 50 mm. The samples were placed on pads so that the height of the samples was minimal (prisms and cylinders are placed on the side). The water temperature in the container was (20 ± 2) °C. Samples were weighed every 24 h of water absorption on ordinary or hydrostatic scales with an accuracy of no more than 0.1%. When weighing on ordinary scales, samples were taken out of the water and pre-wiped with a squeezed damp cloth. The mass of water that leaked from the pores of the sample on the scale cup was included in the mass of the saturated sample. The test was carried out until the results of two consecutive weighings differed by no more than 0.1%.

The water absorption of each concrete sample WM (% by mass) was calculated with an uncertainty of up to 0.1%, using Equation (7):(7)$$W_{M}=\frac{M_{d}-M_{s}}{M_{d}}\cdot 100\%$$
where *M_s_* is the mass of the water-saturated sample (g), and *M_d_* is the mass of the dry sample (g).

After obtaining the water absorption data (Table 6), the dependence of the effect of modifiers and volumetric fiber reinforcement on the water absorption of concrete was defined.

### 2.5. Determination of the Average Density

Tests to determine the average density of concrete were conducted according to Russian State Standard GOST 10181–2014—Concrete Mixtures; methods of testing are available online at https://docs.cntd.ru/document/1200115733 (accessed on 26 February 2022) [36]—where the volume of samples of regular shape was calculated by their geometric dimensions. The dimensions of the samples were determined using a ruler or caliper with an error of no more than 1%. The weight of the samples was determined by weighing, with an accuracy of no more than 0.1%.

The average density *ρ**_W_* (kg/m^3^) of concrete of each sample in the series was calculated with an error of up to 1 kg/m^3^ using Equation (8):(8)$$\rho_{W}=\frac{m}{V}\cdot 1000,$$
where *m* is the mass of the sample (kg), and *V* is the volume of the sample (m^3^). The average density of concrete was calculated as the arithmetic mean value of the test results of all series samples.

After obtaining the average density data (Table 6), the effects of modifiers and volumetric fiber reinforcement on the average density of the concrete were defined.

## 3. Results and Discussion

The summary in Table 3 reflecting concrete compositions of popular classes shows that the use of 10% reactive–active modifier CCM and a high-performance hyperplasticizer reduces the consumption of binder and mixing water without losing the characteristics of workability of the concrete mixture.

Figure 2 shows the modifier’s effect (CCM) on cement consumption.

The diagram shows that the use of modifiers allows cement consumption to be reduced while maintaining the workability of the mixture. Thus, for compositions No. 2 and No. 3, cement consumption was reduced by 14.1%, 17.1%, 19.5%, and 15.4% for concretes B25, B30, B35, and B40, respectively, while for composition No. 4, cement consumption was reduced by 12.8%, 15.9%, 18.4%, and 14.3%, respectively.

The use of a polycarboxylate-ester-based hyperplasticizer reduced the W/C ratio of the compositions (Figure 3), thereby increasing the early and final strength of the concrete while maintaining a given workability, according to the basic law of concrete strength. Figure 3b shows how the W/C ratio decreases as the amount of HPC in the concrete mixture increases. The equation expressed can be used to optimize the W/C ratio in relation to the amount of HPC. The use of HPC allows the concrete mixture to be placed as densely as possible at optimal water and cement consumption, enabling the highest concrete density and strength values for a given grade and cement consumption.

The next stage of the study was laboratory tests of the selected compositions to determine their physical, technical, and operational properties (Table 6).

Table 6 shows that the composition No. 2 of concrete with active silica showed higher values of density, frost resistance, and volumetric water absorption compared with the reference composition No. 1 of concrete without modifiers, indicating higher characteristics of the operational reliability of products made of concrete with active microsilica. This testifies in favor of the theory of creating additional centers by linking hydrolyzed lime with an additional amount of strong cementitious calcium hydrosilicates, which improve the structure of the cement stone. The indicators of concrete composition No. 3 with active microsilica and hyperplasticizer show the highest compressive strength values compared to the others. This is consistent with the basic law of concrete strength, where the optimization of water consumption leads to an increase in the compressive strength of cement concrete. The performance of concrete composition No. 4 with the addition of polypropylene fiber showed better flexural strength characteristics compared to compositions without fiber. This clearly shows the validity of the theory of the necessity of volumetric fiber reinforcement of cement concrete with low-modular polypropylene fiber in order to improve its cracking resistance and to increase concrete resistance to bending loads. The results of the study, in addition to their introduction to the plant for the production of reinforced concrete Firma-BENT LLP (Almaty, Kazakhstan), could be applied to other operating factories for the production of reinforced concrete in Kazakhstan to improve the quality and characteristics of the products. It should be noted that this work did not consider the quantitative indicators of the optimal values of active silica, hyperplasticizer, and polypropylene fiber. However, future research by the authors will be aimed at determining these quantitative parameters and their impact on the physical and technical characteristics of concrete.

Figure 4, Figure 5, Figure 6 and Figure 7 show the dependences of flexural strength, density, volumetric water absorption, and frost resistance of concrete on the application of modifiers and volumetric fiber reinforcement.

Figure 4 shows that the use of polypropylene fibers in the concrete composition No. 4 increased the flexural strength by more than 30% compared with the concrete compositions No. 1, No. 2, and No. 3, which did not use fibers.

According to Figure 5, we can observe an increase in the concrete density of compositions No. 2 and No. 3 involving modifiers, compared with the reference composition No. 1. However, the density values of composition No. 4, involving fiber, are slightly different from those of the reference compositions that do not contain additives. It is assumed that this is due to the effect of air involvement in the concrete mixture when using fibers.

In Figure 6, we can clearly see the decrease in the volumetric water absorption index of composition No. 2 containing microsilica and composition No. 3 containing both microsilica and hyperplasticizer. This corresponds well with the theory of compaction of cement stone and the creation of additional centers of crystallization due to the reaction of active microsilica and the formation of crystalline calcium hydrosilicates.

The test results of the modified concrete in Figure 7 show a general trend of increasing the index of frost resistance from W200 for concretes without additives to W300 for modified concretes of the same grades. Moreover, indicators of volumetric water absorption decreased by up to 40%, which indicates the compliance of the obtained results with the theoretical assumptions about the creation of additional crystallization centers and the reduction in pore space in the concrete body when using reactive pozzolanic additives (active SiO_2_ microsilica).

The strength results show that the obtained characteristics correspond to the normative values. Thus, concretes with modifying additives and volumetric fiber reinforcement show more than 30% increase in flexural strength, which is a highly significant indicator, as the improvement of deformation properties and cracking resistance of concrete by using dispersed reinforcement with polypropylene fiber was the ultimate task of this study.

Comparing the obtained results with the achievements of previous studies [38,39,40], the following can be noted: The volumetric fiber reinforcement in cement concrete with low-modulus polypropylene fiber increases the flexural strength similarly to the structures made of nanocomposite reinforced with carbon nanorubber. However, cement-based concretes have found more widespread use in the global construction industry due to their low cost, ease of use, and availability of raw materials.

## 4. Conclusions

Based on numerous laboratory and industrial tests, it can be concluded that the use of modified concretes is justified both economically and scientifically. Unlike ordinary compositions of concrete, modified concretes allow significant savings of cement (up to 100 kg per 1 m^3^), with better performance in terms of density, frost resistance, and volumetric water absorption, which indicate the best operating characteristics. Thus, when producing concrete sleepers with B40 graded concrete, the application of concrete mix modified with microsilica as 10% of the binder allows savings of up to 70 kg of cement per 1 m^3^, i.e., to reduce the consumption of expensive cement CEM I 52.5 N per 1 m^3^ from 455 kg to 385 kg, while maintaining all strength characteristics as specified in Russian State Standard GOST 10181–2014—Concrete Mixtures; methods of testing available online: https://docs.cntd.ru/document/1200115733 (accessed on 26 February 2022) [37]. 

Application of hyperplasticizers based on polycarboxylates is justified by several positive effects at once, such as water reduction—i.e., reduction in W/C ratio—and correspondingly increasing strength characteristics according to the basic law of concrete strength, improving the physical properties of concrete, increasing density, and reducing permeability, which follows from indicators of volumetric water absorption, while also improving the surface quality of products.

One of the main strength indicators of concrete and a characteristic sought in implementing these studies is the flexural strength, as the main purpose of introducing fibers is to improve the characteristics of concrete under flexural loads. Thus, according to the study results, it can be concluded that the use of low-modulus polypropylene fibers in optimal quantities is justified in terms of improving the weakest characteristic of concrete—flexural strength [24].

## 5. Recommendations and Applications

From the above conclusions, the use of modifiers and dispersal reinforcement can be recommended to reduce the cost and improve the deformation and operational properties of heavy concrete.

Volumetric fiber reinforcement improves the physical–technical and deformation characteristics of existing reinforced concrete mixes, in particular increasing the flexural strength by over 30%. Volumetric fiber reinforcement with low-modular polypropylene fiber of concrete mixtures can be recommended to produce reinforced concrete products.

The effect of using low-modulus plastic fiber in the frames of high-rise buildings is that it increases the flexural strength which, consequently, reduces the consumption of steel reinforcement and the cross-section of concrete partitions. Accordingly, all of these measures reduce the load on the foundation of a high-rise building.

After obtaining positive results from laboratory tests, production testing of the developed compositions was carried out at the plant to manufacture reinforced concrete Firma-BENT LLP (Almaty, Kazakhstan). After confirming the effectiveness of the compositions and real savings of binder in production conditions, the reinforced concrete plant included the developed compositions of concrete in its technological maps. These compositions are recommended for use in their working program to produce reinforced concrete products.

## Figures and Tables

**Figure 1 materials-15-02648-f001:**
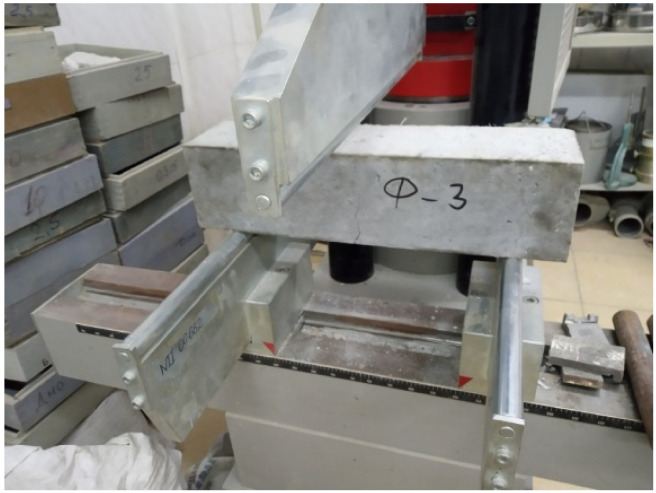
Beam flexural test.

**Figure 2 materials-15-02648-f002:**
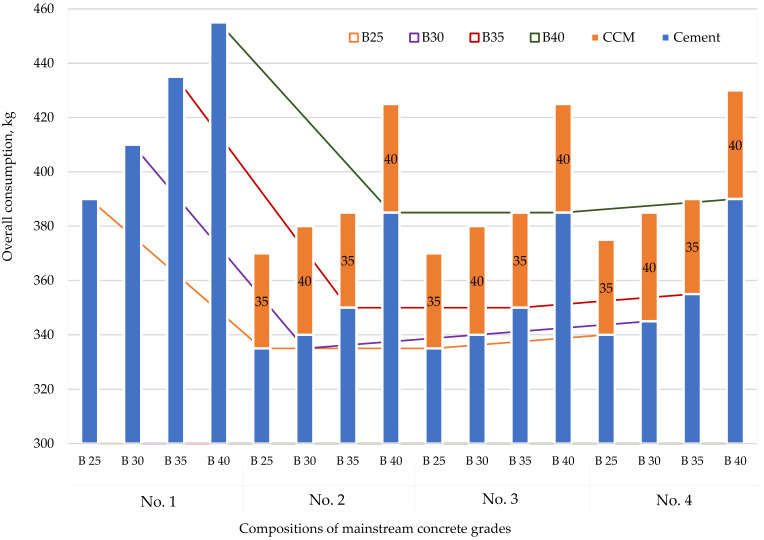
Effect of the modifier on cement consumption: No. 1—no additives; No. 2—with CCM; No. 3—with CCM and HPC; No. 4—with CCM, HPC, and fiber.

**Figure 3 materials-15-02648-f003:**
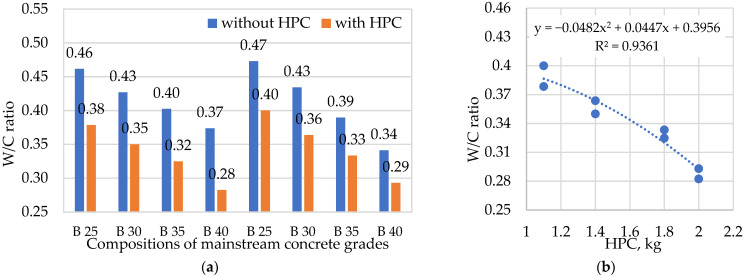
Effect of HPC on W/C ratio: (**a**) dependence of W/C on HPC; (**b**) dependence function.

**Figure 4 materials-15-02648-f004:**
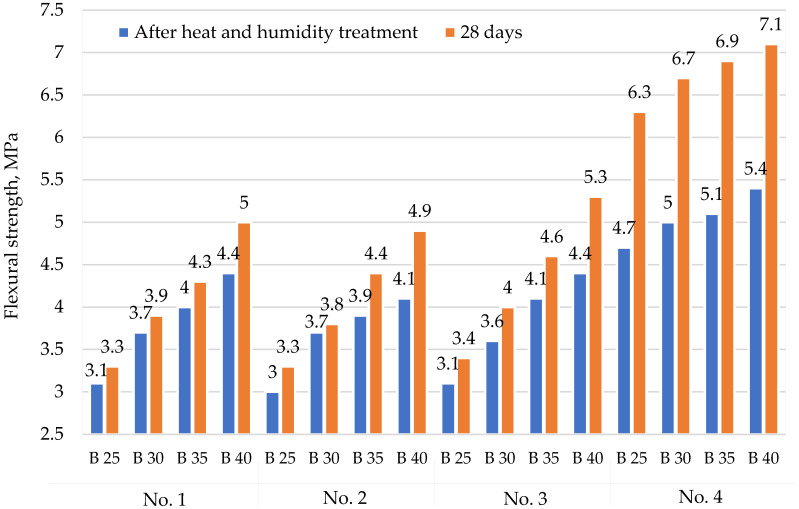
Dependence of flexural strength on the application of modifiers and volumetric fiber reinforcement: No. 1—no additives; No. 2—with CCM; No. 3—with CCM and HPC; No. 4—with CCM, HPC, and fiber.

**Figure 5 materials-15-02648-f005:**
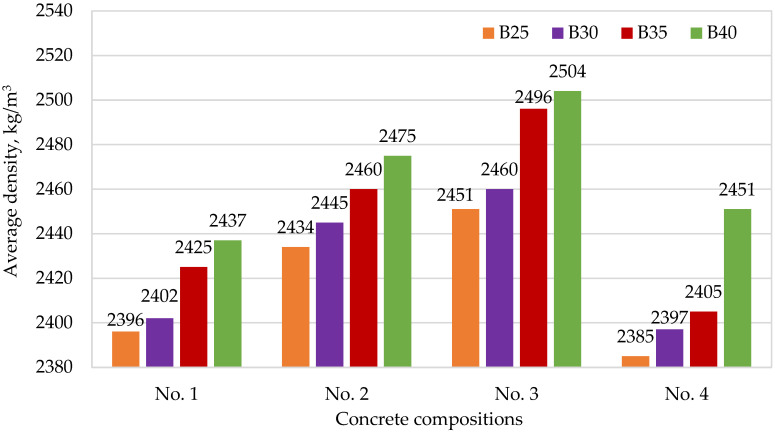
Dependence of concrete density on the application of modifiers and volumetric fiber reinforcement: No. 1—no additives; No. 2—with CCM; No. 3—with CCM and HPC; No. 4—with CCM, HPC, and fiber.

**Figure 6 materials-15-02648-f006:**
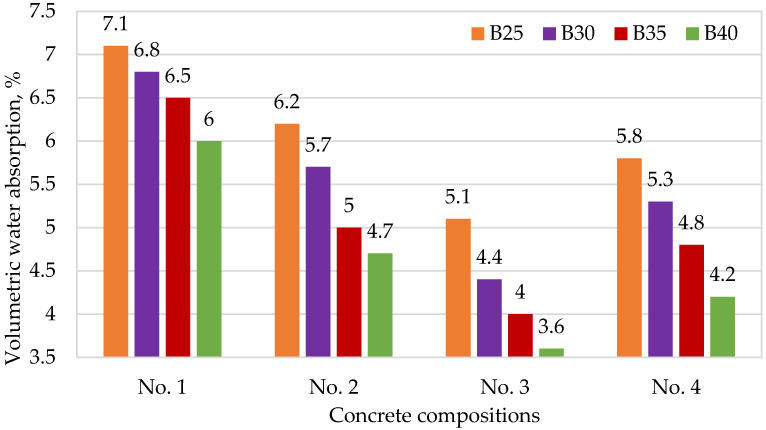
Dependence of volumetric water absorption of concrete on the application of modifiers and volumetric fiber reinforcement: No. 1—no additives; No. 2—with CCM; No. 3—with CCM and HPC; No. 4—with CCM, HPC, and fiber.

**Figure 7 materials-15-02648-f007:**
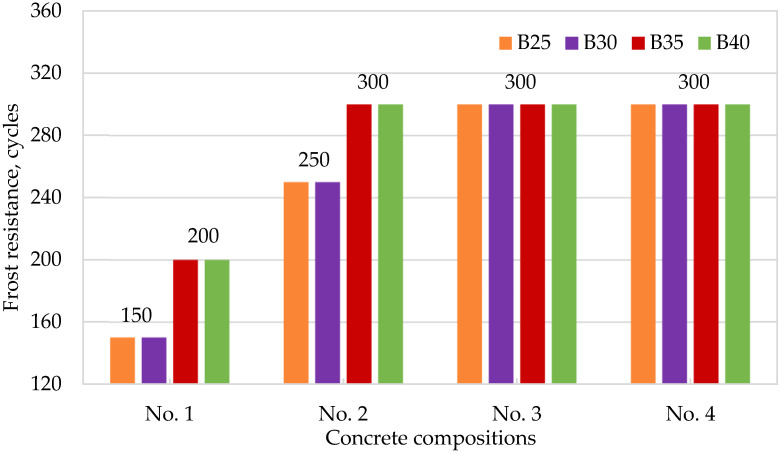
Dependence of concrete frost resistance on the application of modifiers and volumetric fiber reinforcement: No. 1—no additives; No. 2—with CCM; No. 3—with CCM and HPC; No. 4—with CCM, HPC, and fiber.

**Table 1 materials-15-02648-t001:** Technical specifications of AR Premium AH hyperplasticizer.

Parameter	Value
Appearance	Homogeneous liquid of light-yellow color
Density at 25 °C (kg/m^3^)	1030–1070
Hydrogen index (pH)	4
Maximum content of Cl ions	0.1

**Table 2 materials-15-02648-t002:** Technical characteristics of polypropylene fiber.

Parameter	Normative Value	Actual Value
Chemical consistency	Polypropylene	Polypropylene
Type	Monofiber	Monofiber
Fiber length	6–15 mm	Compliant
Fiber diameter	18–21 microns	20 microns
Shape	Round	Round
Surface	Finished with a special compound that facilitates dispersion and adhesion to cement mortar	Finished with Silastol CUT 70
Density	0.91 g/cm³	0.91 g/cm³
Fiber frequency	19.8 mln/kg	Compliant
Tensile strength	320–600 MPa	Compliant
Young’s modulus	3500–3900 MPa	Compliant

**Table 3 materials-15-02648-t003:** Consumption of materials per 1 m^3^ of concrete mixture.

Composition No.	Design Grade	Cement CEM I 52.5 N (kg)	Fiber (kg)	Microsilica (kg)	Hyperplasticizer (kg)	Crushed Stone (kg)	Sand (kg)	Water (L)	Total Consumption (kg)
1	B 25	390	-	-	-	1000	850	180	2420
	B 30	410	-	-	-	1030	810	175	2425
	B 35	435	-	-	-	1100	780	175	2490
	B 40	455	-	-	-	1150	700	170	2475
2	B 25	335	-	35	-	1100	770	175	2415
	B 30	340	-	40	-	1140	740	165	2425
	B 35	350	-	35	-	1180	710	150	2425
	B 40	385	-	40	-	1200	685	145	2455
3	B 25	335	-	35	1.1	1100	770	140	2380
	B 30	340	-	40	1.4	1140	740	133	2393
	B 35	350	-	35	1.8	1180	710	125	2400
	B 40	385	-	40	2.0	1200	685	120	2430
4	B 25	340	3	35	1.1	1095	770	150	2393
	B 30	345	3	40	1.4	1135	740	140	2403
	B 35	355	3	35	1.8	1175	751	130	2449
	B 40	390	3	40	2.0	1195	685	126	2439

Note: workability of the concrete mixture P1.

**Table 4 materials-15-02648-t004:** Test conditions for determining the frost resistance [8].

Method, Grade of Frost Resistance	Test Conditions			Concrete Type
	Saturation Environment	Environment and Freezing Temperature	Environment and Thawing Temperature	
Basic methods				
First, F1	Water	Air, −18 °C	Water, (20 ± 2) °C	All types, except for road and airfield concretes and structures operating under the influence of saline water.
Second, F2	5% aqueous NaCl mixture	Air, −18 °C	5% aqueous NaCl mixture, (20 ± 2) °C	Concretes of road and airfield pavements and concretes of structures operating under the influence of saline water
Accelerated methods				
Second	5% aqueous NaCl mixture	Air, −18 °C	5% aqueous NaCl mixture, (20 ± 2) °C	All types, except for road and airfield pavement concretes and structures operating under the influence of saline water, as well as lightweight concretes with an average density of less than D1500.
Third	5% aqueous NaCl mixture	5% aqueous NaCl mixture, minus (50 ± 2) °C	5% aqueous NaCl mixture, (20 ± 2) °C	All types of concrete, except for lightweight concretes with an average density of less than D1500.

**Table 5 materials-15-02648-t005:** Test conditions applied.

Sample Size (mm)	Freezing		Thawing	
	Minimum Time (h)	Temperature (°C)	Minimum Time (h)	Temperature (°C)
100 × 100 × 100	2.5	−18	2 ± 0.5	20 ± 2
150 × 150 × 150	3.5		3 ± 0.5	

**Table 6 materials-15-02648-t006:** Physical and mechanical properties of concrete compositions.

Composition No.	Design Grade	Compressive Strength (MPa) (*R_b_*)		Flexural Strength (MPa) (*R_bt_*)		Frost Resistance, Cycles (F)	Volumetric Water Absorption (%)	Average Density (kg/m^3^)
		After Heating	28 Days	After Heating	28 Days			
1	B 25	24.9	32.4	3.1	3.3	150	7.1	2396
	B 30	29.7	38.7	3.7	3.9	150	6.8	2402
	B 35	31.4	45.3	4.0	4.3	200	6.5	2425
	B 40	37.2	52.7	4.4	5.0	200	6.0	2437
2	B 25	25.2	32.6	3.0	3.3	250	6.2	2434
	B 30	30.1	38.9	3.7	3.8	250	5.7	2445
	B 35	35.1	45.5	3.9	4.4	300	5.0	2460
	B 40	42.0	53.1	4.1	4.9	300	4.7	2475
3	B 25	26.0	32.5	3.1	3.4	300	5.1	2451
	B 30	29.3	39.0	3.6	4.0	300	4.4	2460
	B 35	39.3	45.1	4.1	4.6	300	4.0	2496
	B 40	47.2	53.5	4.4	5.3	300	3.6	2504
4	B 25	24.6	33.0	4.7	6.3	300	5.8	2385
	B 30	29.5	39.7	5.0	6.7	300	5.3	2397
	B 35	36.2	50.1	5.1	6.9	300	4.8	2405
	B 40	44.1	52.5	5.4	7.1	300	4.2	2451

## Data Availability

The data presented in this study cannot be shared at this time. They may be available from the first author upon reasonable request.

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
