# Peer review of "The Effect of Low-Modulus Plastic Fiber on the Physical and Technical Characteristics of Modified Heavy Concretes Based on Polycarboxylates and Microsilica"

_materials, 2022, doi:10.3390/ma15072648_

Round 1

Reviewer 1 Report

ollowing are the suggestions for the authors.

  • Please use the short keywords for the paper.
  • Author use different abbreviation at different places, which confused the reader, Please provide the list of the abbreviation, please use in the start.
  • The introduction needs to be more emphasized on the research work with a detailed explanation of the whole process considering past, present and future scope.
  • All figures are well plotted however please include more discussion, especially for Figure 3b. Please discuss the HPC effect on the mechanical properties of the concrete.
  • Research gaps should be highlighted more clearly and future applications of this study should be added. Please make the link between the advances images used in this paper with the material properties.
  • What are the effect of low-modulus plastic fiber on the physical and technical characteristics of modified heavy concretes on the framing of the high-rise building regarding the partition loads.?
  • The conclusion part is a little bit prolix and should be improved. It seems to be a sum of results. Also, limitations, future scope and recommendations of this study are suggested to be written as a separate section.
  • Please kindly make revision on the language of the paper presentation. There are still some minor typos and grammatical errors.

Author Response

Dear Reviewer,

Thank you for reviewing our paper, we really appreciate it. We made major changes according to your comments and suggestions. Please consider below our response to each of them (Please see the attachment). 

Best regards,

Authors

Reviewer 2 Report

The problem is timely and interesting. I recommend the publication of the manuscript after the following revisions are properly made:

1) The aim of the paper is not completely well-specified. The authors could specify more this aspect in the abstract and in the introduction of the manuscript.

2) Please, check if all the symbols used in the various equations are well-defined in the text.

3) English should be enhanced throughout the manuscript to eliminate grammatical errors and misprints.

4) For general readers, authors are encouraged to discuss other kinds of structures such as: [(a) "Static stability analysis of carbon nanotube reinforced polymeric composite doubly curved micro-shell panels”; (b) “Effect of nonlinear FG-CNT distribution on mechanical properties of functionally graded nano-composite beam”; (c) “On the mechanics of nanocomposites reinforced by wavy/defected/aggregated nanotubes”].

5) Figs. 6 and 7 should be more discussed.

6) In conclusion, give only main findings of your research with an appropriate value.

Author Response

(The authors gave the same response as above.)

Reviewer 3 Report

The article is interesting and well-organized. I do not see any substantive errors in it. The drawings could be clearer. And the cited bibliography is more contemporary. 

Author Response

Dear Reviewer,
Thank you for reviewing our paper, we really appreciate your positive feedback. We made the necessary changes according to your comments and suggestions. Please consider below our response to each of them (Please see the attachment).
Best regards,
Authors

Reviewer 4 Report

General Notes

Many different methods and standards are mentioned throughout the paper, I think the paper may benefit from brief summaries of these, if this is possible. 

Abstract

  • Line 31: Suggest changing to “…when using both microsilica and superplasticizer
  • Line 33: Suggest adding “successfully introduced” if that statement is accurate

Introduction

  • Line 45: suggest changing to “carbon dioxide (CO2) occur during…”
  • Line 48: suggest removing “taking into account which they operate” and changing to “used both when selecting…”
  • Line 51: I would consider combining this paragraph with the prior paragraph. I’d also consider breaking the sentence beginning on line 53 into multiple smaller sentences; as written it makes sense but requires multiple reads to understand.
  • Line 65: Suggest clarifying what “One of which” refers to, I assume chemical modifiers?
  • Line 74: suggest changing “contribute” to “contributing”
  • Line 76: Is this reduction a numerical reduction or are authors referring to chemical reduction? Please specify.
  • Line 81: Suggest changing to “exert both chemical and physical-chemical…”
  • Line(s) 73, 79, 84: Paragraphs beginning on these lines all discuss additives and their effects and could probably be combined for clarity
  • Line 90: Starting a paragraph with “Besides” is awkward, suggest omitting “Besides”
  • Line 93: suggest changing to “cement concrete of higher flexural strength”
  • Line 97: The sentence beginning on line 97 could be combined with the sentence following it on line 98
  • Line 103: suggest changing to “concrete with highly desirable characteristics but also…”
  • Line 106: the bulleted entries to this list don’t need to end in semicolons and a period for the last entry

Materials and Methods

  • Line 148: again I’m not sure punctuation at the end of each bullet point is necessary
  • Line 156: this sentence seems like it is formatted oddly, like it’s not part of the following paragraph
  • Line 157: insert a space between Heidelberg and Cement
  • Line 160: suggest changing to “The methods specified in the standard are used to determine…”
  • Line 161: I don’t know that a bulleted list is appropriate for this data, it would be easier to read as a normal paragraph, maybe with testing results displayed within parenthesis after each test performed
  • Line 176: suggest changing to “necessary to use sand with dust inclusions less than 1.5%”
  • Line 179: Suggest stating which size fraction relates to 2.6 coarseness
  • Line 205: see notes for lines 106 and 148
  • Equations 1,2,3: Suggest removing commas at the end of each equation
  • Line 234: are these lines meant to be a caption(s) below equations 4 and 5?
  • Line 277: Suggest changing to “samples were frozen in the air, saturated with sodium chloride and thawed in a mixture of sodium chloride.”
  • Line 279: suggest changing mode to keep consistent throughout the paper, perhaps method or conditions

Results and Discussion

  • Line 331: suggest changing “reduced to” to “reduced by”
  • Line 332: same suggestion as line 331
  • Figure 5: It is interesting that the addition of fiber to the CCM and HPC mix caused the resulting concrete to drop from the densest to least dense. Is this solely due to the relatively low density of the fibers added? Or are there other effects occurring?
  • Figure 7: suggest changing name to figure 7
  • Figure 7: Not sure why a 3-d plot was selected for this data when there are only 2 axes used, suggest displaying as bar graphs like others

Conclusion

  • Line 373: “it can conclude” should be changed to “it can be concluded”
  • Line 385: suggest changing to “is justified by several positive characteristics…”

Author Response

Dear Reviewer,
Thank you for reviewing our paper, we really appreciate it. It helped really improve the English of the paper. We made the necessary changes according to your comments and suggestions. Please consider below our response to each of them (Please see the attachment).
Best regards,
Authors
